# Maternal Dietary Diversity and Birth Weight in Offspring: Evidence from a Chinese Population-Based Study

**DOI:** 10.3390/ijerph20043228

**Published:** 2023-02-12

**Authors:** Yuxin Teng, Hui Jing, Samuel Chacha, Ziping Wang, Yan Huang, Jiaomei Yang, Hong Yan, Shaonong Dang

**Affiliations:** Department of Epidemiology and Biostatistics, School of Public Health, Xi’an Jiaotong University Health Science Center, Xi’an 710061, China

**Keywords:** dietary diversity, birth weight, health, score, pregnancy, newborn

## Abstract

Studies on the association between maternal dietary diversity and birth weight in offspring are limited, and the impact of such an adjustable factor on birth weight requires investigation to promote neonatal health. This study used data from a larger-scale population-based survey conducted in northwest China to evaluate the association of maternal dietary diversity with neonatal birth weight with a generalized estimating equation model. The results found that maternal dietary diversity was positively associated with neonate birth weight. Furthermore, a higher minimum dietary diversity for women (MDD-W) score during pregnancy was related to a lower risk of low birth weight (LBW) in offspring. The mothers with the highest MDD-W score had a 38% (OR = 0.62, 95% CI: 0.43–0.89) lower risk of LBW than those with the lowest score. Similarly, the mothers with the highest animal-based food dietary diversity score (DDS) had 39% (OR = 0.61, 95% CI: 0.38–0.98) lower risk of LBW in offspring compared with those with the lowest animal-based food DDS. Moreover, the ratio of animal-based food DDS to non-animal-based food DDS could play an important role in predicting neonate birth weight. In conclusion, increasing maternal dietary diversity would improve birth weight in offspring, especially by increasing the intake of animal-based foods among the Chinese population.

## 1. Introduction

Birth weight is an important evaluation indicator of fetal development in utero during pregnancy [1], and it is closely related to neonatal health, growth and development [2]. Globally, low birth weight (LBW) has been estimated to account for 15% to 20% of live births [3]. LBW is twice as common in developing countries as in developed ones, and the incidence of LBW was found to be 4.59% in China based on the latest nutritional health survey of children and pregnant women [2,4]. As an adverse pregnancy outcome, LBW is highly associated with infant mortality, especially in neonates, in the whole world [5,6]. Therefore, identifying modifiable risk factors for adverse birth outcomes, such as LBW, is essential to improving child health outcomes and has important implications for public health, especially for the disadvantaged child population [7,8].

The dietary intake of pregnant women should meet not only their own nutritional needs but also the needs of fetal growth [9,10]. The first 1000 days of life (pregnancy to 24 months after birth) is regarded as a window of opportunity to promote healthy child development, to which maternal nutrition is largely attributed [11]. Malnutrition during pregnancy is associated with adverse pregnancy outcomes, including intrauterine growth restriction, which greatly increases the risk of preterm birth, LBW, stunted growth, and neonatal death [12,13]. Studies showed that a high-quality maternal diet could prevent adverse birth outcomes, such as LBW, preterm birth, and small-for-gestational-age (SGA) birth weight [8]. Therefore, ensuring adequate nutritional intake during pregnancy is of great significance for maternal health and normal fetal development [14]. Several studies have shown that dietary diversity is positively correlated with nutrient intake and its sufficiency [15]. People are advised to improve the nutritional quality of their diet by increasing dietary diversity [16].

The dietary diversity score (DDS) is an evaluation index of dietary quality that assesses the diversity of all foods consumed in a given period, based on a simple count of food groups consumed [17]. The Food and Agriculture Organization (FAO) proposes the use of the minimum dietary diversity for women (MDD-W) as a tool to measure the dietary diversity of women [18,19]. MDD-W is a 10-item food-based dietary diversity indicator that is usually used in the measurement of nutrient adequacy among women living in resource-poor areas and developing countries [20], but now it is also used to assess maternal dietary diversity [21]. Because diet is a modifiable factor, it is of great significance in investigating the association between maternal dietary diversity and birth outcomes. Unfortunately, the impact of maternal dietary diversity on improving adverse pregnancy outcomes remains unclear [22]. Limited evidence came mainly from developing countries. A study in Tanzania reported that MDD-W was inversely associated with SGA, and diet quality was inversely associated with very low birth weight, LBW, fetal loss, and very preterm birth [8]. The FAO Women’s Dietary Diversity Scores of ≥4 food groups during pregnancy were shown to be associated with lower risk of maternal anemia, LBW, and preterm birth in rural Ethiopia [11]. Long-term anemia and insufficient intake of vegetables (dark green leafy vegetables), dairy, and fruits were associated with higher risk of experiencing at least one adverse pregnancy outcome [23]. A small number of studies from resource-rich areas showed that increasing dietary food intake during pregnancy can promote fetal weight, length, and bone development in the American population [24]. In addition, in Spanish women, higher DDS was related to lower SGA risk and lower nutrient deficient intake, probably due to high diet quality [22]. These results suggested that DDS still could play a positive role in improving birth outcomes, even in woman with a relatively rich diet. Thus far, few relevant studies have used MDD-W to explore the association between dietary diversity and birth weight, especially for the Chinese population. Therefore, we conducted a study on maternal dietary diversity and birth weight among Chinese newborns, which would provide essential information on further improving fetal growth by modifying maternal dietary patterns.

## 2. Methods

### 2.1. Data and Participants

Data were from a population-based cross-sectional study conducted in Shaanxi of Northwest China. The detailed study design has been reported elsewhere [25,26]. Briefly, the women who were pregnant during the period of 2010–2013 and had pregnancy outcomes before the survey were enrolled. Women were excluded if they could not answer the survey because of limited cognitive capacity. A stratified multistage random sampling method was used to obtain the sample. A structured questionnaire was used to collect information on sociodemographic characteristics, lifestyle, antenatal examination, and reproductive history of women and birth outcomes of offspring by face-to-face interview. All investigators received the same strict pre-investigation training. The training included language expression, ways of asking questions, unified handling of answers to questions with ambiguous attitudes, and so on. All information were collected from throughout the regions of Shaanxi by using standard Chinese or a dialect if necessary. In total, 30,027 women consented to participate in the survey. Among the participants, 8209 women with young children aged less than 2 years were asked to report their diets during pregnancy. In the present study, we excluded 342 women with incomplete maternal dietary information, 803 women with an implausible dietary energy intake (less than 4500 KJ/day or greater than 20,000 KJ/day) [27], 147 women with twin or multiple pregnancies, and 112 women without offspring birth weight, leaving 6805 eligible participants for the final analysis. The participants included and those excluded were similar, and a slight difference was found only in the characteristics of household expenditure, maternal registered permanent residence, antenatal examination, folic acid supplements, and illness during pregnancy (Appendix A). The study was conducted in accordance with the guidelines stated in the Declaration of Helsinki. All procedures were approved by the Xi’an Jiaotong University Health Science Center (No. 2012008), and all participants signed informed written consent.

### 2.2. Maternal Dietary Intake

The mothers were asked to recall food intake when they were pregnant with the young child. A semi-quantitative food frequency questionnaire (FFQ) was adopted to collect data regarding dietary food intake during the whole pregnancy. This FFQ was based on the previously validated FFQ designed for pregnant women in rural western China [28], and it included 102 foods with 13 staple foods, 19 animal protein foods, 6 non-animal protein foods, 36 vegetables, 12 fruits, 3 nuts, 5 drinks, and 8 snacks. The intake frequency and portion size (g per one-time) of each food was recorded, and the portion sizes for each food were recorded with the assistance of food portion images [29]. The frequency of food intake was divided into eight categories as almost or no intake, <1 time/month, 1–3 times/month, 1 time/week, 2–4 times/week, 5–6 times/week, 1 time/day, and ≥2 times/day.

### 2.3. Measures of Maternal Dietary Diversity

MDD-W, as a dietary evaluation index for women with the minimum dietary diversity, is assessed based on main food categories [18]. In the present study, the foods were divided into 10 categories based on the Chinese Food Guide Pagoda [30], including meat, poultry and fish, nuts and seeds, eggs, dark green leafy vegetables, dairy, pulses, other vegetables, other vitamin-A rich fruits and vegetables, other fruits, and grains, which was consistent with previous studies [31].

To prevent bias resulting from negligible consumption in a food group [32], the revised scoring method was used in the present study. If the intake of any food in a given food group met the daily intake of more than 25 g and was consumed at least once a week, one point was allocated to this food group and zero point otherwise. The scores of all food groups were summed as MDD-W score ranging from 0 to 10, and a greater score meant a higher maternal dietary diversity. Furthermore, according to the food source, the food groups were divided into animal-based and non-animal-based food groups. Animal-based foods contained four food groups, namely meat, poultry and fish, dairy, and eggs. Non-animal-based foods included grains, pulses, nuts and seeds, dark green leafy vegetables, other vitamin-A rich fruits and vegetables, other vegetables, and other fruits. The specific DDS of two such food groups were estimated, and the ratio of animal-based food DDS to non-animal-based food DDS was established.

### 2.4. Assessment of Neonate Birth Weight

According to the child’s birth certificate index in the survey, the gestational age, neonate birth weight, birth date, and gender were extracted. Birth weight was measured within 1 h of delivery using a baby scale to the nearest 10 g by the maternal and child health care staff in the local hospitals at or above the township. Gestational age was calculated from the last menstrual period or confirmed by ultrasound scan. Low birth weight was defined as neonatal birth weight < 2500 g [33].

### 2.5. Ascertainment of Covariates

According to the previous studies [8,34,35,36], potential covariates included in the present study were classified mainly into two parts. One potential covariate was sociodemographic characteristics including maternal age (≤20 years, >20 & ≤30 years, or >30 years), gestational weeks (continuous), family economic status (low, middle, or high), maternal registered permanent residence (rural or urban), maternal education level (low, middle, or high), maternal career (farmers, workers and merchants, or intellectuals), and neonate gender (male or female). Another potential covariate was about health-related behaviors during pregnancy including antenatal examination level (low, middle, or high), illness during pregnancy (yes or no), and folic acid supplements (yes or no).

Family economic status, a comprehensive index, was established with principal component analysis (PCA) based on average monthly household expenditure, house condition, and vehicle ownership, which represented family economic level and was categorized as low, middle, or high based on tertiles [37]. A comprehensive assessment score representing antenatal examination level was also established with PCA based on four variables including month of initial antenatal examination, antenatal examination, genetic heredity counseling, and antenatal diagnosis, and the score was classified equally into three categories as low, middle, or high. For folic acid supplementation, users were defined as those who used folic acid supplements alone or folic acid-containing multivitamins before or during pregnancy; otherwise, the subjects were classified as non-users. Illnesses during pregnancy included common illnesses such as colds and fevers.

Because there were some values missing for maternal age (*n* = 63), gestational weeks (*n* = 14), neonate gender (*n* = 97), average monthly household expenditure (*n* = 455), home ownership (*n* = 127), vehicle ownership (*n* = 127), maternal registered permanent residence (*n* = 38), initial antenatal examination (*n* = 145), antenatal examination (*n* = 24), genetic heredity counseling (*n* = 79), antenatal diagnosis (*n* = 72), folic acid supplements (*n* = 31), illness during pregnancy (*n* = 17), maternal education level (*n* = 17), and maternal career (*n* = 51), the method of multiple imputations was adopted to impute the missing values [38].

### 2.6. Statistical Analysis

The characteristics of the participants were expressed as mean ± standard deviation or percentage. The MDD-W score was classified into three categories according to tertiles (T1/T2/T3). Differences in the characteristics of women across three MDD-W groups were compared using ANOVA for continuous variables and Chi-square test for categorical variables. The proportion of mothers with an MDD-W score of one point and a mean MDD-W score for each food group was presented between LBW and non-LBW group.

Considering that the participants were possibly clustered in counties of rural areas or districts of urban areas, the generalized estimating equation (GEE) models were used to explore the association between maternal dietary diversity and birth weight considering cluster effect. The GEE model is a population average model used to estimate the associations between health outcomes and neighborhood characteristics (factors which are relevant to the neighborhoods to which the study individuals belong and which possess both physical and social attributes that may plausibly affect the health of individuals) in multilevel studies [39]. In this study, the variable of district/county was regarded as the random effect, and pregnancy maternal dietary diversity with the birth weights of different newborns and covariates had fixed effects in the GEE models. The unadjusted and adjusted models were established to estimate the change in birth weight with normal distribution and identity link function and OR for LBW with binomial distribution and log link function, as well as their accompanying 95% CI. Considering the influence of covariates, three incremental models were established to explore the robustness of association of interest. Model 1 was the unadjusted model without covariates. Model 2 was adjusted for maternal age, gestational weeks, neonate gender, and folic acid supplements. Model 3 was further adjusted for family economic status, maternal registered permanent residence, antenatal examination level, illness during pregnancy, maternal education level, and maternal career based on model 2.

We also conducted subgroup analysis by main covariates to explore potential associations of interest across the subgroups. Furthermore, we completed a series of sensitivity analyses to examine the robustness of the association of interest. First, a quantile regression was used to investigate the impact of maternal dietary diversity across the whole distribution of birth weight. Second, considering the contribution of each food group to dietary diversity, the principal component analysis method was adopted to establish alternative dietary diversity score (DDS-PCA), which was used to explore whether the association of interest would change with different assessment methods for DDS. Third, the analysis was restricted to the participants with children aged less than 12 months in order to reduce the effect of recall bias. Last, the analysis was repeated in the participants with complete values of all original covariates. All analyses were performed using the SPSS 18.0 and R 4.2.1, and statistical significance was defined as *p* values < 0.05.

## 3. Results

### 3.1. The Characteristics of the Participants

In total, 6805 newborns and their mothers were included in this study. The mean age of mothers was 26.13 ± 4.63 years old, and the mean MDD-W score was 6.11 ± 1.97. The mean gestational week at the time of birth was 39.56 ± 1.33, and the mean birth weight of the newborn was 3267.79 ± 449.32 g. Compared to those with the lowest score of MDD-W (T1), mothers with the highest score (T3) tended to have a high family economic status, have a high maternal education level, be intellectual, have high antenatal examination level, live in urban areas, and take folic acid supplements during pregnancy. The neonate birth weight ascended with increases in MDD-W score (*p* = 0.007) (Table 1).

### 3.2. Maternal Dietary Diversity between LBW and Non-LBW Groups

The MDD-W score in LBW was significantly lower than that in non-LBW groups (5.82 ± 2.06 vs. 6.12 ± 1.97) (*p* = 0.023). Figure 1 indicates the dietary diversity distribution and mean scores based on MDD-W for each food group. The proportion of mothers with an MDD-W score of one point was about 38.1% for egg and 52.0% for dairy food in LBW group, which was lower than 46.8% and 63.9% in non-LBW group (the left side of Figure 1). Moreover, a significant difference was observed in the mean MDD-W score for egg and dairy food groups between the LBW and the non-LBW group (the right side of Figure 1) (*p* < 0.05).

### 3.3. Association between Maternal Dietary Diversity and Birth Weight or Low Birth Weight

Table 2 shows that the neonate birth weight was significantly related to maternal overall MDD-W score and animal-based food DDS after adjusting for all potential covariates. When overall MDD-W score increased by one unit, the neonate birth weight increased by 5.37 g (95% CI: 0.07, 10.66). Similarly, the neonate birth weight increased by 11.40 g per one-unit increase in animal-based food DDS (95% CI: 0.47, 22.32). The neonate birth weight seemed to increase by 22.48 g per one-unit increase in ratio, although it was not statistically significant (95% CI: −16.26, 61.21).

Logistic regression analysis indicated that the neonate’s low birth weight was significantly associated with maternal overall MDD-W score, animal-based food DDS, and the ratio after adjusting for all potential covariates (Table 3). When overall MDD-W score increased by one unit, the risk of LBW was reduced by 9% (OR: 0.91, 95% CI: 0.85, 0.98). Similarly, the risk of LBW was reduced by 27% per one-unit increase of animal-based food DDS (OR: 0.73, 95% CI: 0.62, 0.86). Moreover, the risk of LBW was reduced significantly by 71% per one-unit increase in the ratio of animal-based food DDS to non-animal-based food DDS (OR: 0.29, 95% CI: 0.14, 0.61). Furthermore, compared with those in T1, the risk of LBW in offspring decreased by 38% among the mothers whose overall MDD-W score was in T3 (OR = 0.62, 95% CI: 0.43, 0.89). Similarly, the risk of LBW in offspring was reduced by 39% for the mothers whose animal-based food DDS was in T3 (OR: 0.61, 95% CI: 0.38, 0.98). However, there was no association found between low birth weight and non-animal-based food DDS.

### 3.4. Subgroup Analysis and Sensitivity Analysis

The direction of the association of maternal dietary diversity with LBW was consistent across the subgroups, implying a stable association of interest. Additionally, the reduced risk of LBW for MDD-W was observed in young mothers aged 20–30 years, rural mothers, mothers taking folic acid supplements during pregnancy, mothers with low or middle antenatal examination levels, and mothers who were farmers by occupation (*p* < 0.05) (Appendix A). To examine the robustness of the association of interest, sensitivity analyses were performed. A quantile regression analysis presented a significant association between maternal MDD-W score and birth weight which was in the middle percentile (q = 0.5, q = 0.6). The higher the score, the higher the birth weight. There was a significant association between the ratio of animal-based food DDS to non-animal-based food DDS and birth weight which was in the lower percentile of birth weight (q = 0.2, q = 0.3) (Appendix A). When DDS-PCA was used for the measurement of maternal dietary diversity instead of MDD-W score, a lower risk of LBW in offspring was also associated with higher DDS-PCA, which was consistent with the results from MDD-W (Appendix A). When analysis was restricted to the participants with a child aged less than 12 months, the results were still in line with those from overall participants (Appendix A). Similar results were also observed in a repeated analysis in the participants with complete values of all covariates (Appendix A).

## 4. Discussion

The present study provided a shred of additional evidence from the Chinese population that maternal dietary diversity was positively associated with neonate birth weight, even after adjustment for potential covariates. A higher MDD-W score was related to a lower risk of LBW. Compared with those in the lowest tertiles, the risk of LBW in offspring decreased by 38% among the mothers in the highest tertiles of overall MDD-W score. A more significant association could be found between animal-based food DDS and neonate birth weight. Besides MDD-W score, the ratio of animal-based food DDS to non-animal-based food DDS could be an important predictor for neonate birth weight. A series of sensitivity analyses further confirmed the association of interest.

The main findings in the present study were similar to previous studies, but the effect size varied [40,41]. Saaka et al. found that higher maternal dietary diversity was associated with a 57% lower risk of LBW in Ghana based on the individual dietary diversity score [41]. In the present study, the higher maternal dietary diversity was associated with a 9% and 27% lower risk of LBW in the Chinese population based on the overall MDD-W score and animal-based food DDS, respectively. Cuco’G et al. found that the neonate birth weight would increase by 7.8 to 11.4g if every 1g of protein was consumed by pregnant women at 10 and 26 weeks of gestation [24]. The present study also found an increased neonate birth weight of 5.37 g for a one-unit increase in overall MDD-W score. Additionally, the neonate birth weight increased by 11.40 g when animal-based food DDS increased by one unit. Such varying effect size could result from differences in the population or measurements of maternal dietary diversity. This present study used MDD-W, an index validated for micronutrient adequacy which has not been used in previous studies [11,41]. Although a higher dietary diversity of vegetables and fruits has also been found to negatively correlate with SGA [31,42], there was no significant association with LBW in the present study. This may be due to easy access to fruits and vegetables in under-resourced areas; as a result, dietary diversity indicated achieving nutrient adequacy among undernourished pregnant women [22]. Among the important findings in the present study was the ratio of animal-based food DDS to non-animal-based food DDS that was found positively associated with neonate birth weight. The risk of LBW was reduced significantly, that is, by 71% per one-unit increase in the ratio, which suggested importance of animal food for maternal and child health. This ratio represented a pattern of maternal dietary diversity to some extent, and a large ratio meant an increase in intake of animal foods, and accordingly, it may partly indicate no significant association of non-animal-based food DDS with neonate birth weight. Another important finding was the association between MDD-W score and neonate birth weight, which could be modified by some sociodemographic characteristics of mothers to some extent. The significant association between MDD-W score and neonate birth weight could imply that dietary diversity of younger mothers should be taken into consideration. Women living in rural areas could have limited access to a healthy dietary pattern [22] and inadequate knowledge about dietary nutrition during pregnancy compared with urban women; thus improvement of their dietary diversity is more necessary. Moreover, a significant association was observed among the participants taking folic acid supplements during pregnancy and those with low or middle antenatal examination levels. As mentioned in previous research, DDS might perform better in severely under-resourced areas where dietary diversity indicates achieving energy and nutrient adequacy among undernourished young children and women of reproductive age [31]. However, it is noteworthy that maternal education level and family income did not significantly modify the association of interest in the present study, which implied that an improvement in dietary pattern might play an important role in the promotion of the dietary diversity of Chinese women regardless of the greater access to individual healthy foods.

Some studies have highlighted that low quality and high inflammatory potential maternal diet was associated with lower offspring birth size and a higher risk of offspring being born small-for-gestational-age [7]. However, little is known regarding the role of maternal DDS in predicting poor outcomes of pregnancy such as LBW. At different stages of fetal growth and development in the uterus, the fetus obtains nutrition from the mother through absorption, nutrition transmission, and other ways and discharges its metabolites through the mother [14]. Therefore, the nutritional status of the mother undoubtedly plays a vital role in the growth and development of the fetus [14]. Promotion of maternal dietary diversity could increase comprehensive intake of all kinds of nutrients during pregnancy to a great extent. Furthermore, the present study suggested a positive relationship between animal-based food intake during pregnancy and neonatal birth weight. This was consistent with other previous findings, in which a higher dietary protein intake, especially animal protein and dairy protein, was associated with higher birth weight and lower risks of SGA and LBW among Chinese pregnant women with low protein intake [43]. Maybe this trend was because the high-quality protein mainly from animal-based food not only participates in the composition of fetal tissues and organs but also provides energy for the mother [14,44]. Moreover, it can provide certain minerals and micronutrients to promote fetal growth and development [45]. In our participants, the difference in maternal dietary diversity between mothers with and without LBW offspring was found mainly in animal food groups such as eggs, dairy, meat, poultry, and fish. This implied that inadequate intake of animal protein and inappropriate dietary pattern could still be a major dietary nutrition problem in pregnant Chinese women. However, a study from developed countries found that there was still a risk of neonatal SGA and maternal nutritional inadequacies. A possible reason might be more access to a healthy dietary pattern for pregnant women, but overconsumption did not guarantee nutritional adequacy [22]. In China, animal foods, especially eggs and dairy, are easy to access [46], and the promotion of maternal dietary diversity represents optimization of dietary pattern. Therefore, it is essential to increase dietary diversity in order to improve neonate birth weight by modifying maternal dietary patterns. It is suggested that the intake of foods with high-quality protein, including eggs, dairy meat, poultry, and fish should be appropriately increased during pregnancy, along with improvement of overall dietary diversity.

The present study has several strengths. First, to our knowledge, it was the first study with a large sample to investigate the association between maternal dietary diversity and neonate birth weight among the Chinese population. Second, our study filled a gap in the research systematically analyzing MDD-W scores during pregnancy and their effect on neonatal birth weight. Third, in a manner distinct from existing studies in other countries, the MDD-W score in the present study was established based on the FAO’s recommendation and the Chinese Food Guide Pagoda. Thus, this index could be more appropriate for reflecting the dietary diversity of pregnant Chinese women. Fourth, previous data-driven dietary patterns have certain limitations in practical guidance, but MDD-W in the present study was more practical and operational. Distinct from previous studies, we limited the minimum food intake when including food groups to prevent overestimation of the results. Moreover, we further used principal component analysis to construct a new dietary diversity score (DDS-PCA) to improve the traditional MDD-W scoring method considering the influence of the biological contribution of diverse foods to diets. However, some limitations have to be addressed carefully when interpreting the results. First, the inference of causality is limited due to the nature of the cross-sectional design. Second, although we controlled for a series of covariates such as maternal age, folic acid supplements, and family economic status in the analysis, there were still some unobserved or unknown confounders that we could not fully investigate. For example, there was lack of maternal gestational weight gain which was related to reduced risks of LBW and SGA. Moreover, the survey was conducted from August to November. Taking into account the long pregnancy period of the participants, there was no special consideration for the influence of seasonal factors. Third, participants were from a regional survey in China, which partly limits the generalization of the results to the general Chinese population. Thus, results of the present study should be interpreted with caution. Fourth, the FFQ has a tendency of overestimating food intake [43], for which we cannot rule out possible misclassification due to recall bias. To reduce errors, we tried to help mothers provide accurate responses in the survey. We also used detailed auxiliary materials such as food portion images and calendars and standard questionnaires to collect information. Prior to the formal survey, a pilot study was conducted to test the survey instruments, and rigorous training was conducted for the interviewers according to the detailed guides. On the other hand, FFQ used in the present study primarily collected participants’ intake of individual foods and could not distinguish food mixing issues. Moreover, we did not collect food insecurity information. Fifth, the traditional MDD-W method could be flawed and did not capture sufficiently the biological contribution of diverse foods to diets [47]. The overall MDD-W score was the sum of the dietary diversity scores of ten food groups, and the influence of the contribution degree of a single food on the scores of the food groups was ignored. Thus, it may weaken the strength of the association between dietary diversity and neonate birth weight. Moreover, it does not consider the weighted impact of each food group on the overall score. Consequently, we further used DDS-PCA for repeated analysis and found that the association between DDS-PCA and neonate birth weight seemed to be more obvious. Last but not least, this study lacks an investigation into the physiological aspects of women, for example, anemia, triglycerides, glycaemia, etc. The need for further research is suggested to conduct a subsequent investigation to examine the correlation between maternal dietary diversity scores, newborn weight, and maternal blood chemical parameters.

Based on our findings, there are some practical implications of this study. Firstly, to health managers, the improvement of dietary diversity during pregnancy, especially by increasing the intake of eggs and dairy food, which is effective in the prevention and control of neonatal low birth weight should be taken into consideration. Maternal dietary nutrition management policies and measures should be amended by considering dietary diversity. Secondly, for maternal and child health workers, MDD-W is easy to understand and operate. In practice, the dietary quality of pregnant women can be generally understood by calculating the MDD-W score, so as to provide guidance for improving the diet during pregnancy. Thirdly, for pregnant mothers, the improvement of dietary nutrition during pregnancy should not only focus on the supplementation of specific food groups such as meat but also promote dietary diversity by intake of multiple food groups.

## 5. Conclusions

In conclusion, increasing maternal dietary diversity, especially by increasing the intake of animal-based foods, would improve birth weight in offspring among the Chinese population. As a modifiable dietary factor, MDD-W could be used as a predictor in maternal dietary practice to improve maternal and children’s health. However, follow-up or intervention studies are needed to further determine this association and the mechanism behind it.

## Figures and Tables

**Figure 1 ijerph-20-03228-f001:**
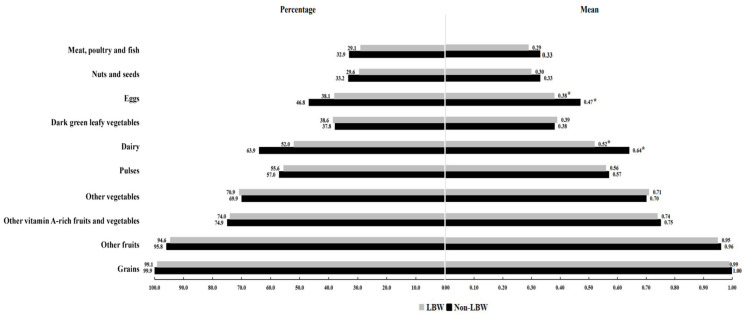
Dietary diversity distribution and mean scores based on MDD-W for each food group. Note: * Comparison between LBW and non-LBW group in mean MDD-W score of such food group: *p* < 0.05.

**Table 1 ijerph-20-03228-t001:** The characteristics of the participants according to tertiles of MDD-W score.

	MDD-W			*p*
	T_1_ (Lowest) *n* = 2594	T_2_ *n* = 2447	T_3_ (Highest) *n* = 1764	
Maternal age (year), $\bar{x}\pm s$	26.0 ± 4.8	26.1 ± 4.6	26.3 ± 4.4	0.131
Gestational weeks (week), $\bar{x}\pm s$	39.6 ± 1.3	39.5 ± 1.3	39.5 ± 1.3	0.071
Neonate birth weight (g), $\bar{x}\pm s$	3248.1 ± 461.6	3271.6 ± 443.5	3291.4 ± 438.0	0.007
Neonate gender, n (%)				
male	1356 (52.3)	1313 (53.7)	958 (54.3)	0.379
female	1238 (47.7)	1134 (46.3)	806 (45.7)	
Family economic status, n (%) *				
low	904 (34.8)	809 (33.1)	550 (31.2)	<0.001
middle	938 (36.2)	852 (34.8)	578 (32.8)	
high	752 (29.0)	786 (32.1)	636 (36.1)	
Maternal registered permanent residence, n (%)				
urban	282 (10.9)	376 (15.4)	337 (19.1)	<0.001
rural	2313 (89.1)	2071 (84.6)	1427 (80.9)	
Antenatal examination level, n (%) **				
low	924 (35.6)	724 (29.6)	396 (22.4)	<0.001
middle	912 (35.2)	796 (32.5)	518 (29.4)	
high	758 (29.2)	927 (37.9)	850 (48.2)	
Folic acid supplements, n (%)				
yes	1815 (70.0)	1837 (75.1)	1378 (78.1)	<0.001
no	779 (30.0)	610 (24.9)	386 (21.9)	
Illness during pregnancy, n (%)				
yes	1532 (59.1)	1433 (58.6)	1040 (59.0)	0.933
no	1062 (40.9)	1014 (41.4)	724 (41.0)	
Maternal education level, n (%)				
low	306 (11.8)	207 (8.5)	107 (6.1)	<0.001
middle	1536 (59.2)	1323 (54.1)	825 (46.8)	
high	752 (29.0)	917 (37.5)	832 (47.2)	
Maternal career, n (%)				
farmers	2005 (77.3)	1737 (71.0)	1162 (65.9)	<0.001
workers and merchants	309 (11.9)	341 (13.9)	283 (16.0)	
intellectuals	280 (10.8)	369 (15.1)	319 (18.1)	

Note: The characteristics of the participants were expressed as mean ± standard deviation or percentage. * Family economic status was established with PCA based on average monthly household expenditure, house condition, and vehicle ownership. ** Antenatal examination level was established with PCA based on month of initial antenatal examination, antenatal examination, genetic heredity counseling, and antenatal diagnosis.

**Table 2 ijerph-20-03228-t002:** Association between MDD-W score and neonatal birth weight by linear regression (*n* = 6805).

	β	95% CI	*p*
Model 1			
Overall MDD-W	8.55	3.07, 14.02	0.002
Animal-based food	20.30	9.15, 31.44	<0.001
Non-animal-based food	6.71	−0.56, 13.97	0.070
Ratio	50.69	10.52, 90.85	0.013
Model 2			
Overall MDD-W	9.15	3.92, 14.38	0.001
Animal-based food	21.12	10.45, 31.80	<0.001
Non-animal-based food	7.45	0.51, 14.40	0.035
Ratio	49.54	10.95, 88.12	0.012
Model 3			
Overall MDD-W	5.37	0.07, 10.66	0.047
Animal-based food	11.40	0.47, 22.32	0.041
Non-animal-based food	4.77	−2.16, 11.71	0.177
Ratio	22.48	−16.26, 61.21	0.255

Note: Model 1 was unadjusted. Model 2 was adjusted for maternal age, gestational weeks, neonate gender, and folic acid supplements. Model 3 was further adjusted for family economic status, maternal registered permanent residence, antenatal examination level, illness during pregnancy, maternal education level, and maternal career. Ratio: ratio of animal-based food DDS to non-animal-based food DDS.

**Table 3 ijerph-20-03228-t003:** Association between MDD-W and neonatal low birth weight by Logistic regression (OR (95% CI)) (*n* = 6805).

	MDD-W		*p*	MDD-W Tertiles			*p* for Trend
	OR	95% CI		T1	T2	T3	
Model 1							
Overall MDD-W	0.93	0.86, 0.99	0.029	1.00	0.65 (0.48, 0.88)	0.64 (0.46, 0.91)	0.007
Animal-based food	0.77	0.66, 0.89	<0.001	1.00	0.61 (0.45, 0.84)	0.62 (0.41, 0.95)	0.003
Non-animal-based food	0.97	0.89, 1.06	0.538	1.00	0.96 (0.69, 1.35)	0.93 (0.68, 1.27)	0.638
Ratio	0.33	0.17, 0.63	0.001	1.00	0.86 (0.63, 1.16)	0.56 (0.39, 0.80)	0.001
Model 2							
Overall MDD-W	0.91	0.85, 0.98	0.014	1.00	0.61 (0.44, 0.85)	0.61 (0.43, 0.89)	0.005
Animal-based food	0.73	0.62, 0.86	<0.001	1.00	0.57 (0.41, 0.79)	0.60 (0.38–0.95)	0.002
Non-animal-based food	0.96	0.87, 1.06	0.406	1.00	0.88 (0.62, 1.25)	0.90 (0.64, 1.25)	0.493
Ratio	0.28	0.14, 0.59	0.001	1.00	0.82 (0.60, 1.13)	0.53 (0.36, 0.77)	0.001
Model 3							
Overall MDD-W	0.91	0.85, 0.98	0.015	1.00	0.60 (0.43, 0.83)	0.62 (0.43, 0.89)	0.005
Animal-based food	0.73	0.62, 0.86	<0.001	1.00	0.57 (0.41, 0.79)	0.61 (0.38, 0.98)	0.003
Non-animal-based food	0.96	0.87, 1.06	0.381	1.00	0.87 (0.61, 1.23)	0.89 (0.64, 1.24)	0.470
Ratio	0.29	0.14, 0.61	0.001	1.00	0.83 (0.60, 1.14)	0.54 (0.37, 0.79)	0.001

Note: Model 1 was unadjusted. Model 2 was adjusted for maternal age, gestational weeks, neonate gender, and folic acid supplements. Model 3 was further adjusted for family economic status, maternal registered permanent residence, antenatal examination level, illness during pregnancy, maternal education level, and maternal career. Ratio: ratio of animal-based food DDS to non-animal-based food DDS.

## Data Availability

Data are available from the authors upon request.

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
