# Peer review of "Maternal Dietary Diversity and Birth Weight in Offspring: Evidence from a Chinese Population-Based Study"

_ijerph, 2023, doi:10.3390/ijerph20043228_

Round 1
Reviewer 1 Report
In my opinion, this is a fascinating paper. Nutrition during pregnancy is vital to the excellent development of the newborn. I believe that the results of this work will impact the Chinese government's decision to design projects for its population to improve nutrition in women with poor access to food quality.
· The main question addressed by the research was to collect information concerning maternal dietary diversity and newborn weights in the Chinese population. In addition, the authors mention the importance of having background information about the diets mothers take during their pregnancies. This information will help to perform a diet plan for Chinese women who have more food deficiencies during their pregnancies.
· I consider this work an original topic that will help to focus on the diet of women of reproductive age.
· Therefore, the only thing that I suggest to the authors for a subsequent investigation is to include a blood chemical test to inspect the physiological aspects of the women, for example, anemia, triglycerides, glycemia, etc. Also, I suggest a correlation test between maternal dietary diversity scores, newborn weight, and mother blood chemical parameters.
· In figure 1, I suggest increasing the font size
· The conclusions are consistent with the evidence and arguments presented. The references are appropriate.
· My recommendation is accepted in the present form
· The English language and style are fine/minor spell check requires
Author Response
请参阅附件。

Reviewer 2 Report
Data from a larger-scale population-based survey conducted in northwest China evaluated the association of maternal dietary diversity with neonatal birth weight. Additional details are lacking in the methods section to show the rigor of the study. The manuscript requires the assistance of an English language review to improve the paper’s clarity.
Considering the study was conducted in 2010-2013, why have the authors now decided to conduct the current analysis?
Who collected the dietary data and was the same language used throughout the regions?
Was food insecurity information collected?
Was the season collected and accounted for in the data collection?
How were mixed dishes handled for the MDD-W?
Who measured the neonatal birth weight?
Footnotes for Table 1 are needed to define how the data is presented in the table.
Figure 1 includes asterisks, and these are not defined.
It is still not clear within the discussion how this study is novel or what gaps it addresses.
Have other studies also analyzed the data as tertiles and how do their findings compare to your study?
Based on the study findings would stakeholders from the community benefit from using any of the tools to obtain data on the local populations and how can the data be used for interventions or programs?
Round 2
Reviewer 2 Report
An additional review of language is needed for minor issues with grammar.
To confirm the information regarding who collected the dietary data, it currently states that they are investigators. Were they indeed investigators or study personnel?
